Methods

# PASCAR: a multiscale framework to explore the design space of constitutive and inducible CAR T cells

Harshana Rajakaruna[1], Milie Desai[2], Jayajit Das[1,3,4]

**CAR T cells are engineered to bind and destroy tumor cells by targeting overexpressed surface antigens. However, healthy cells expressing lower abundances of these antigens can also be lysed by CAR T cells. Various CAR T cell designs increase tumor cell elimination, whereas reducing damage to healthy cells. However, these efforts are costly and labor-intensive, constraining systematic exploration of potential hypotheses. We develop a protein abundance structured population dynamic model for CAR T cells (PASCAR), a framework that combines multiscale population dynamic models and multi-objective optimization approaches with data from cytometry and cytotoxicity assays to systematically explore the design space of constitutive and tunable CAR T cells. PASCAR can quantitatively describe in vitro and in vivo results for constitutive and inducible CAR T cells and can successfully predict experiments outside the training data. Our exploration of the CAR design space reveals that optimal CAR affinities in the intermediate range of dissociation constants effectively reduce healthy cell lysis, whereas maintaining high tumor cell-killing rates. Furthermore, our modeling offers guidance for optimizing CAR expressions in synthetic notch CAR T cells. PASCAR can be extended to other CAR immune cells.**

## Introduction

Chimeric antigen receptor (CAR) T cells have been widely successful in treating hematologic cancers (Lim and June, 2017; Ellis et al, 2021). CAR T cells are engineered to express CAR molecules which bind to antigens overexpressed in tumor cells and stimulate cytotoxic T cell responses. However, healthy cells can express such antigens at low copy numbers (Yasui et al, 1988; Klichinsky et al, 2020) and can become targets of CAR T cell cytotoxicity (Morgan et al, 2010; Hernandez-Lopez et al, 2021; Duan et al, 2022). Such off-target destruction to healthy tissues is a major source of severe immune-related adverse effects in patients undergoing CAR T cell therapy (Parkhurst et al, 2011; Guo et al, 2018) and can become a

major issue for solid tumors where viral organs are damaged by CAR T cells. Therefore, generating an optimal CAR T cell response that maximizes tumor cell elimination, whereas minimizing off-target destruction has been a longstanding pursuit of CAR T cell therapies. A wide range of design strategies to engineer CAR T cells have been proposed, including constitutive co-expression multiple CARs (Srivastava & Riddell, 2015; Cho et al, 2018) that sense several target antigens overexpressed by cancer cells or generation of inducible CAR expressions that tune copy numbers or abundances of CARs (Roybal et al, 2016; Hernandez-Lopez et al, 2021) depending on the abundances of target antigens present on target cells. However, most of these design strategies are conceived intuitively, and thus limit systematic and wide explorations of the design space for CAR T cells.

Optimal design strategies in economics or engineering often involve optimization of conflicting objectives where the best trade-offs between objectives are explored by subjecting computational/mathematical models to multiple objective optimization or Pareto optimization (Atashkari et al, 2005; Deb, 2011; Dworczak et al, 2021). However, the application of similar approaches for designing CAR T cells face a major challenge because of the lack of experimentally validated and computationally efficient multiscale mathematical/computational models that can describe CAR T cell responses against target cells. Though a large variety of population dynamic models based on ordinary differential equations (ODEs) (Chaudhury et al, 2020) have been developed to describe kinetics of populations of CAR T cells interacting with cancer cells in vitro or in vivo, most of these models do not explicitly include CAR ligand affinities, CAR abundances, co-receptors, and their cognate ligand molecules or biochemical signaling processes initiated upon engagement of CARs with cognate ligands. Because the design of CAR constructs often involves manipulation of the CAR affinities, CAR abundances or T cell signaling processes, it is difficult to systematically explore the design space of CAR constructs with such existing models.

CAR abundances in single CAR T cells can vary over 1,000-fold in CAR T cell populations (Hernandez-Lopez et al, 2021) which could play an important role in regulating the response of CAR T cell

[1]The Steve and Cindy Rasmussen Institute for Genomics, The Abigail Wexner Research Institute, Nationwide Children's Hospital, Columbus, OH, USA [2]Department of Biology, Indian Institute of Science Education and Research, Pune, India [3]Department of Pediatrics and Pelotonia Institute for Immuno-Oncology, College of Medicine, Columbus, OH, USA [4]Biophysics Program, The Ohio State University, Columbus, OH, USA

Correspondence: jayajit@gmail.com

populations against target cells. For example, in vitro cytotoxic assays of T cells with inducible CAR expressions showed that an increase in mean CAR abundances by less than 1.3-fold can produce over threefold increase in the rate of lysis of target cells in vitro. Furthermore, abundances of target antigens (e.g., CD19) on tumor cells in patients can also vary over 100-fold and lead to disparate (responders or nonresponders) outcomes in patients undergoing CAR T cell therapy (Majzner et al, 2020). A handful of recent pharmacodynamic models (Singh et al, 2020; Qi et al, 2022) incorporated CAR ligand interactions by considering mean abundances of CARs and their cognate ligands, however, these models are unable to capture the variations of single-cell CAR abundances or the T cell signaling kinetics. Detailed agent-based models incorporating details of receptor–ligand interactions, T cell signaling kinetics and cell metabolism have been developed to describe CAR T cell responses (Prybutok et al, 2022), however, quantitative validation of these models with experiments is challenging because of the presence of many difficult-to-calibrate model parameters and computationally intensive nature of the simulations.

We develop an experimentally validated protein abundance structured population dynamic model for CAR T cells (PASCAR) that integrates molecular receptor–ligand interactions to single CAR T cell signaling and activation to population kinetics of interacting CAR T cells and target cells. PASCAR integrates CAR–ligand interactions and the ensuing signaling kinetics with a minimal but generalizable mechanistic modeling approach using ODEs where model parameters can be well estimated from routinely carried out experiments such as cytotoxicity assays and flow cytometry. We demonstrate PASCAR can quantitatively describe in vitro results for constitutive and inducible CAR T cells and can successfully predict experiments outside the training data. Constitutive CAR T cells constitutively express CAR molecules, whereas inducible CAR T cells generate CAR expressions depending on their interaction with target antigens on target cells. PASCAR is then combined with a Pareto optimization that includes the trade-off between lysis of tumor and healthy cells to explore the design space of CAR constructs. Our investigations show CAR–ligand affinities with dissociation constants in the micromolar range can dramatically decrease healthy cell lysis but sustain a high rate of tumor cell killing. The proposed framework can be extended to model responses in other CAR immune cells (Klichinsky et al, 2020; Liu et al, 2020; Kararoudi et al, 2022).

## Model development

We developed a protein abundance structured model (PASCAR) for describing interacting populations of CAR CD8[+] T cells and target cells. In the model, individual CAR T cells interact with single-target cells where the interaction between a CAR CD8[+] T cell and a target cell is initiated by binding of the CAR with its cognate ligand such as HER-2. Once the CAR–ligand complex is formed, it goes through a series of chemical modifications such as phosphorylation of tyrosine residues in the CAR-associated CD3ζ adaptors because of signaling reactions (Harris et al, 2018; Rohrs et al, 2020). These modifications eventually lead to the release of lytic granules by the CAR T cell which induces disintegration of the target cell membrane and eventual death of the target cell (Davenport et al, 2018). The signaling reactions can also

induce proliferation of the CAR CD8[+] T cells (Sahoo et al, 2020; Faude et al, 2021). There are a multitude of interconnected signaling processes involving many proteins, lipids, and ions that link the formation of the CAR–ligand complex to the lysis of target cells or to CAR T cell proliferation. In the model, we make simplifying assumptions to model these signaling reactions to relate the rate of target cell lysis and CAR CD8[+] T cell proliferation to the abundances of CAR–ligand signaling complexes. Because signaling reactions occur at faster time scales (~minutes) (Huse et al, 2007) compared with the time scales (~hours) of target cell lysis (Weigelin et al, 2021) or CAR T cell proliferation (Obst, 2015), we assume that the rates of target cell lysis and T cell proliferation are influenced by the steady state values of the abundances of CAR–ligand complexes in our model. To simplify the notation, we denote CAR and its cognate ligand by R and H, respectively, and denote the single-cell copy numbers or abundances of these proteins by the italicized versions of the symbols. In PASCAR, a CAR T cell with $R$ number of CARs interacts with a target cell with expressing $H$ number of cognate ligands. R binds with H at a rate $k_{on}$ to create the complex, R–H, which unbinds at a rate $k_{off}$ (Fig 1). The formation of the complex R–H induces a series of signaling reactions in the CAR T cell that eventually leads to the activation of the CAR T cell.

We propose two models, Model NKP and Model KP, to investigate different mechanisms of CAR T cell activation because of CAR binding to cognate ligands. In Model NKP, CAR molecules in CAR T cells interact with cognate ligands on target cells to form a R–H complex and activation of a CAR T cell is assumed to be proportional to the steady state abundance of the R–H complex (Fig 1). In Model KP, we use an approximate model for CAR T cell signal transduction based on McKeithan's kinetic proofreading (KP) model (McKeithan, 1995). In this model, the R–H complex formed in the CAR T cell undergoes $N$ number of modifications representing chemical modifications by downstream signaling reactions to create an active complex $C_N$ (Fig 1). Production of $C_N$ leads to cytotoxic response and proliferation of CAR T cells. The above series of reactions approximate signaling reactions in CAR T cells initiated by the formation of the CAR–ligand complex that produces activation (e.g., phosphorylation) of key signaling proteins (Salter et al, 2018, 2021) such as Zap70 or PLCγ or SLP-76 crucial for CAR T cell activation. The chemical modifications of the R–H complex are described by first-order reactions (Fig 1) with rate $k_p$. At any chemically modified state of the complex, unbinding the ligand leads to immediate reversion of all the modifications in the complex and the complex reverts to the native unbound ligand and receptor state. This step is known as the KP step which creates a sharp increase in the steady state number of $C_N$ as $k_{off}$ decreases. The KP model gives rise to a waiting time ($\tau_w$) that the receptor–ligand complex should last to generate productive signaling—receptor–ligand complexes with lifetimes (~1/$k_{off}$) larger than this waiting time (i.e., 1/$k_{off}$ > $\tau_w$ ∝1/$k_p$) generates active complexes $C_N$ at greater abundances compared with short-lived (1/$k_{off}$ ≪ $\tau_w$) receptor complexes. Below, we describe the kinetics of populations of interacting CAR T cells and target cells for Model NKP and Model KP using ODEs.

### Model NKP
In this model, the rate of CAR T cell proliferation or the rate at which a CAR T cell lyses an interacting target cell is proportional to the steady state abundance of the R–H complex or $C_0$. The abundance

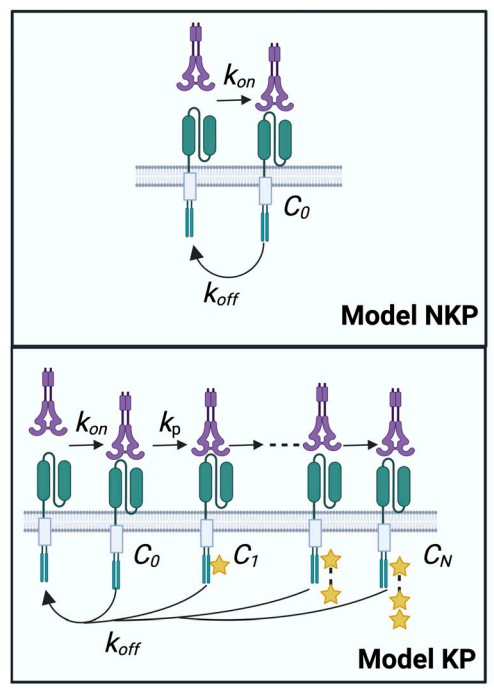

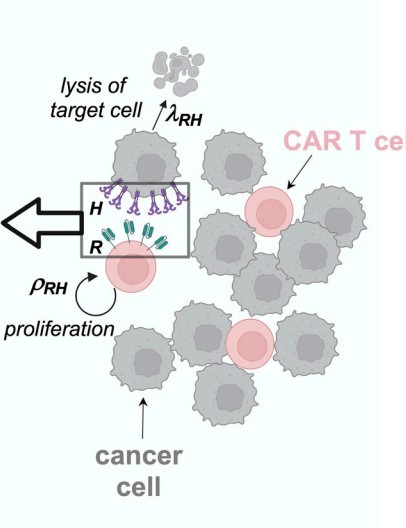

**Figure 1. Multiscale protein abundance structured population dynamic model for CAR T cells (PASCAR) model.**
Single chimeric antigen receptor (CAR) T cells interact with single target cells in the PASCAR model. The strength of CAR T cell signaling depends on the abundance of the CAR–human epidermal growth factor receptor 2 (HER2) complex ($C_0$) in Model NKP or on the abundance of an active complex ($C_N$) formed because of $N$ number of chemical modifications in the CAR–HER2 complex in Model KP. The abundances of $C_0$ (Model NKP) or $C_N$ (Model KP) depend on the abundances of CAR ($R$) and HER2 ($H$) in single CAR T cell and the target cell, respectively. The rate of lysis ($\lambda_{RH}$) of target cells because of the cytotoxic response or the proliferation rate ($\rho_{RH}$) of CAR T cells depends on the abundances of $C_0$ (Model NKP) or $C_N$ (Model KP) in single CAR T cells. The lysis and the proliferation rates are used to describe the kinetics of populations of CAR T cells and target cells.

($C_0$) of the R–H complex formed at the steady state as a function of $R$, $H$, and the dissociation constant $K_D = k_{off}/k_{on}$ is given by

$$C_0(R, H, K_D) = \frac{1}{2}(R + H + K_D)\left(1 - \sqrt{1 - \frac{4RH}{(R + H + K_D)^2}}\right)$$

We assume that the rate of CAR T cell proliferation $\rho_{RH}$ and the rate of target cell lysis $\lambda_{RH}$ are given by

$$\rho_{RH} = \rho_c C_0(R, H, K_D)$$

$$\lambda_{RH} = \lambda_c C_0(R, H, K_D)$$

where $\rho_c$ and $\lambda_c$ are the proportionality constants.

### Model KP

The rates of CAR T cell proliferation and target cell lysis are proportional to the steady state abundance of the end complex $C_N$ which as a function of the steady state abundance of R-H or $C_0^{(KP)}$, $k_p$, and $k_{off}$ is given by,

$$C_N\left(C_0^{(KP)}, k_{off}, k_p\right) = C_0^{(KP)}\beta\left(\frac{k_p}{k_{off} + k_p}\right)^N \quad (1)$$

where $\beta = 1 + \frac{k_p}{k_{off}}$. $C_0^{(KP)}$ is given by,

$$C_0^{(KP)}\left(R, H, K_D, k_p, k_{on}, \beta\right) = \frac{1}{2\beta}(R + H + K_D)\left(1 - \sqrt{1 - \frac{4RH}{(R + H + K_D)^2}}\right) \quad (2)$$

The derivations of Equations (1) and (2) are shown in Supplemental Data 1. The proliferation and lysis rates in Model KP are given by

$$\rho_{RH} = \rho_c C_N\left(C_0^{(KP)}, k_{off}, k_p, N\right)$$

$$\lambda_{RH} = \lambda_c C_N\left(C_0^{(KP)}, k_{off}, k_p, N\right)$$

Now, we set up kinetic equations for a population of target cells interacting with a population of CAR T cells. The target cells can represent healthy or tumor cells. We consider a population of CAR T cells where individual CAR T cells express $R$ copy number of CARs within a range $[R_{min}, R_{max}]$. The CAR T cells interact with a population of target cells where any single-target cell expresses $H$ copy number of cognate ligands within a range $[H_{min}, H_{max}]$. The target cells replicate with a rate $r$ and the target cell population decreases as they are lysed by interacting CAR T cells. The CAR T cells proliferate because of their interaction with the target cells, and the CAR T cells die with a constant rate $\delta$. The population kinetics can be described in terms of the number ($U_H$) of target cells each carrying $H$ copy number of cognate ligands and the number ($T_R$) of CAR T cells each carrying $R$ copy number of CARs and as follows:

$$\frac{dU_H}{dt} = rU_H - \sum_{R' = R_{min}}^{R_{max}} \lambda_{R'H} T_{R'} U_H \quad (3)$$

$$\frac{dT_R}{dt} = \sum_{H' = H_{min}}^{H_{max}} \rho_{RH'} U_{H'} T_R - \delta T_R \quad (4)$$

Given the initial condition $\{T_R(0), U_H(0)\}$, the above system of ODEs describe the composition or the structure of the populations of CAR T cells and target cells at any later time.

**Parameter estimation** The affinity parameters ($k_{on}$, $k_{off}$) are usually measured in in vitro using experiments such as surface

plasmon resonance (Ghorashian et al, 2019) or titrations using flow cytometry (Sharma et al, 2020). The single-cell abundances of CARs and their cognate ligands are measured in quantitative flow cytometry experiments (Hernandez-Lopez et al, 2021) which can be used to estimate $\{T_R (0), U_H(0)\}$. The distributions of R are modeled as log-normal distributions where the mean ($\mu_R (0)$) and the SD ($\sigma_R (0)$) for the distributions at t = 0 are estimated. The signaling parameter $k_p$, the proportionality constants, $\lambda_c$ and $\rho_c$, can be estimated from cytotoxicity assays where CAR T cells and target cells are co-cultured for different CAR T cell and target cell ratios (or E:T ratios) and the fraction of lysed target cells is measured after few days (e.g., 3 d). We used data available from Hernandez-Lopez et al (2021) for constitutive and synthetic notch (synNotch)-CAR T cells to estimate the parameters in our model. Further details are provided in the Materials and Methods section.

**Pareto optimization** There is a trade-off between maximizing lysis of cancer cells and minimizing lysis of healthy cells by CAR T cells. For example, CAR T cells that lyse target cells expressing a small number of cognate ligands can lyse cancer cells efficiently but can also produce large off-target killing of healthy cells. We set up a multi-objective optimization problem to systematically explore the space of optimal parameter values in our PASCAR model and calculate the Pareto front, which represents the set of optimal parameter values where any objective cannot be optimized further without choosing less than optimal values for the other objectives. We set up a two-objective optimization problem where the percentage of lysed healthy cells and the inverse of the percentage of the lysed cancer cells at a fixed time (e.g., 5 d) post co-incubation of the CAR T cells and the target cells are minimized simultaneously. We consider a mixture of healthy and cancer cells where healthy and cancer cells express on average $10^{4.5}$ and $10^{6.5}$ human epidermal growth factor receptor 2 (HER2) molecules/cell (Hernandez-Lopez et al, 2021). The CAR T cells are introduced at t = 0 and interact with the target cells following the PASCAR model, and after a fixed time interval τ, we evaluate the total number of lysed healthy and cancer cells. The parameters in the PASCAR model for constitutive and synNotch-CAR T cells are varied to evaluate the Pareto fronts. Further details of the calculation are provided in the Materials and Methods section.

# Results

### Model with kinetic proofreading captures lysis of target cells by constitutive CAR T cells

We evaluated the capability of Model NKP and Model KP to describe the response of constitutive CAR T cells expressing anti-HER2 single-chain antibody (scFv) against target cells expressing HER-2 in vitro. We fitted both the models to the percentage lysis data obtained from cytotoxicity assays of target cells co-cultured with constitutive CAR T cells and to abundances of CARs in the T cells in the co-culture assayed after 3 d (Hernandez-Lopez et al, 2021). We converted the binding rate $k_{on}$ and the dissociation constant $K_D$ from the units of nM to molecules/cell. The high

($K_D$ = 17.6 nM, $k_{off}$ = 9.0 × $10^{-5}$ $s^{-1}$, $k_{on}$ = 5.1 × $10^3$ $M^{-1}$ $s^{-1}$) and low ($K_D$ = 210 nM, $k_{off}$ = 6.8 × $10^{-4}$ $s^{-1}$, $k_{on}$ = 3.2 × $10^4$ $M^{-1}$ $s^{-1}$) affinitiy CARs used in the experiments by Hernandez-Lopez et al (2021) are converted to the units of $K_D$ = 2.39 molecules/cell, $k_{off}$ = 9.0 × $10^{-5}$ $s^{-1}$, $k_{on}$ = 3.76 × $10^{-5}$ (molecules/cell)$^{-1}$ $s^{-1}$, and $K_D$ = 28.47 molecules/cell, $k_{off}$ = 6.8 × $10^{-4}$ $s^{-1}$, $k_{on}$ = 2.38 × $10^{-5}$ (molecules/cell)$^{-1}$ $s^{-1}$, respectively. The details of the conversion are shown in the Materials and Methods section. We also evaluated the effect of membrane diffusion on estimation of $k_{on}$, $k_{off}$, and $K_D$ in Equation (1) to find that the values are changed by a small amount (≤20%) from their well-mixed counterpart (Supplemental Data 2) which induces negligible changes in the values of $C_N$ (Fig S1). Therefore, we used the rate constants without considering diffusion for simplicity. We found that Model NKP is unable to describe the increase in percentage lysis (Fig 2A) as the affinity of the CAR increases from high ($K_D$ = 17.6 nM, $k_{off}$ = 9.0 × $10^{-5}$ $s^{-1}$) to low ($K_D$ = 210 nM, $k_{off}$ = 6.8 × $10^{-4}$ $s^{-1}$), though the model fits the means and the variances of CAR abundances in CAR T cells at t = 3 d reasonably well (Figs S2A and B). This is because the abundances of CAR–HER2 complexes formed in Model NKP for the high- and low-affinity CARs are roughly similar (Fig 2B) which produces almost equal rates of lysis and proliferation for the CAR T cells bearing the high- and low-affinity CARs. Next, we fitted Model KP to the same percentage lysis and CAR expression data which successfully captured the increase in the lysis by the CAR T cells expressing high-affinity CARs (Fig 2C). In the model, the increase in percentage lysis with increasing HER2 density closely follows the increase in $C_N$ (Fig 2D) with the HER2 density. The model also reasonably fitted the means and variances of the CARs expressed at t = 3 d (Figs 2E and F). In addition, we computed the Akaike Information Criterion (AIC) for the fits for the KP and the NKP models using AIC = n [$ln$ (SSR/n) + $ln$ ($2\pi$) + 1] + 2$k$ where n (=36) is the number of data points, $k$ is the number of model parameters, and SSR is the sum of squared residuals. We used the cost function in Equation (5) to calculate the SSR and $k$ = 4 and 6 for the NKP and the KP models, respectively. The AIC values for the NKP (AIC = 135.66) and the KP (AIC = 112.05) models show that the KP model is favored substantially (ΔAIC ≫ 2 [Burnham et al, 2011]) over the NKP model. The estimated parameters (Table 1) show that inclusion of kinetic proofreading in the signaling kinetics with and active complex formed at N ≈ 7 steps can separate the CAR T cell responses between high and low-affinity CARs for the same HER2 concentrations (Fig 2D). We further used Model KP to describe the percentage lysis of target cells when constitutive CAR T cells expressed high affinity ($K_D$ = 17.6 nM, $k_{off}$ = 9.0 × $10^{-5}$ $s^{-1}$) CARs at higher and lower abundances than that considered above. We fitted the percentage lysis for the higher and lower CAR abundances for the cytotoxicity assay performed at the 1:0.35 E:T ratio to estimate the distributions of CAR abundances at t = 0 and kept all other parameters fixed at the values estimated (Table 1) in previously from low and high-affinity CARs. Next, we predicted the percentage lysis for cytotoxic assays of at E:T ratio 1:1 for target cells expressing different HER2 abundances (Fig 2G). The predictions agreed with data reasonably ($R^2$ ≈ 0.9) (Fig 2H); however, the model systematically underpredicted the percentage lysis by ~20% at higher HER2 abundances. This could point to parameters pertaining to signaling kinetics affected by CAR abundances due

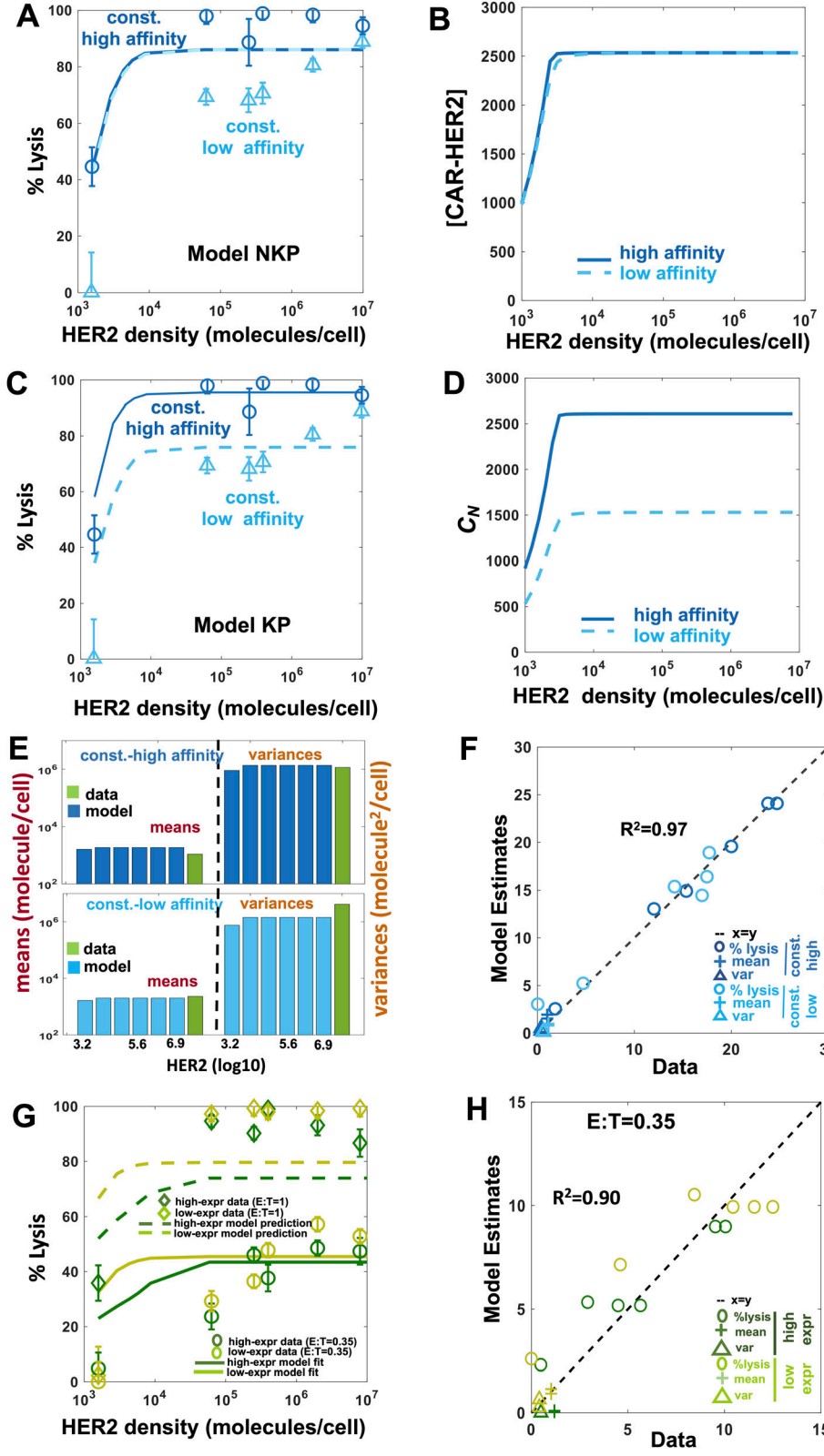

**Figure 2. Protein abundance structured population dynamic model for CAR T cells modeling of cytotoxic and proliferation responses of constitutive chimeric antigen receptor (CAR) T cell against target cells.**
**(A)** Shows fits for Model NKP to the percentage lysis of target cells 3 d after 10,000 constitutive CAR T cells were incubated with 20,000 target cells. The data and the fits are shown for five different human epidermal growth factor receptor 2 (HER2) expressions with average HER2 abundances at $10^{4.7}$, $10^{5.2}$, $10^{5.7}$, $10^{6.2}$, $10^{6.9}$ molecules/cell. The constitutive CAR T cells express either high affinity ($K_D$ = 17.6 nM, $k_{off}$ = 9.0 × $10^{-5}$ s$^{-1}$) or low affinity ($K_D$ = 210 nM, $k_{off}$ = 6.8 × $10^{-4}$ s$^{-1}$) CARs. The values of percentage lysis at HER2 abundances that are in between the values mentioned above were calculated in the model by interpolating the means and variances of the HER2 distributions (see Figs S11A–D and S12A and B for details). **(A, B)** Variation of the CAR ligand abundance (or $C_0$) with the copy numbers of HER2 ligands expressed on target cells for the high- and low-affinity CARs in (A). **(A, C)** Shows fits for Model KP to the percentage lysis data described in (A). **(A, D)** Variation of the active CAR–ligand complex (or $C_N$) with the copy numbers of HER2 ligands expressed on target cells for the high- and low-affinity CARs in (A). **(E)** Comparison of the measured means and the variances of CAR abundances at day 3 post co-incubation with the fits obtained from Model KP. The mean HER2 abundances of the target cells used for modeling the co-culture experiments are shown along the x-axis. The means and variances for CAR abundances obtained from Hernandez-Lopez et al (2021) are shown with the green bar. **(F)** Comparison of the model fits for the percentage lysis, and means and variances of distributions of CAR abundances at day 3 post co-incubation with their experimental counterparts. The variables are made nondimensional by scaling the variables by their respective SDs calculated using the measured values. The goodness of fit is quantified by the correlation coefficient $R^2$ which shows an excellent agreement ($R^2$ = 0.97) between the model and the data. **(G)** Shows fits (solid lines) for Model KP to the percentage lysis data at day 3 for co-culture experiments at E:T = 1:0.35 (20,000 target cells and 7,000 CAR T cells) with constitutive CAR T cells expressing high-affinity CAR ($K_D$ = 1.9 nM, $k_{on}$ = 1.2 × $10^4$ M$^{-1}$ s$^{-1}$, $k_{off}$ = 2.2 × $10^{-5}$ s$^{-1}$) at high (WT) and low (+degron) abundances (Hernandez-Lopez et al, 2021) with HER2 expression distributions given for target cells at different mean HER2 levels. The predictions (dashed lines) for percentage lysis at day 3 generated from Model KP for co-culture experiments at E:T = 1:1 (20,000 target cells and 20,000 CAR T cells) are compared with available measurements in Hernandez-Lopez et al (2021) corresponding percentage lysis co-culture are constitutive model fitted (think line) to day 3 CAR expression data, estimating only the initial CAR (high affinity) distribution (mean and variance) for high (WT)- and low (+degoron)-expression CARs, with all other parameters best estimated by the earlier models fitted to high- and low-affinity CAR for constitutive and synthetic notch CAR. **(G, H)** Shows goodness of the fit for Model KP with the data (percentage lysis, mean, and variances of the CAR abundances at day 3) for the co-culture experiments in (G) for E:T = 1:0.35. The comparison is shown for the nondimensional variables where they are scaled by their SDs calculated using the measured values.

**Table 1. List of estimated and fixed parameter values.**

| Parameter | Unit | Constitutive + synNotch | Constitutive |
|---|---|---|---|
| $\lambda_c$ | day$^{-1}$ | $2.09 \times 10^{-8}$ [$1.78 \times 10^{-8}$, $2.42 \times 10^{-8}$] | $1.33 \times 10^{-8}$ [$7.66 \times 10^{-9}$, $2.05 \times 10^{-8}$] |
| $k_p$ | s$^{-1}$ | 0.0074 [0.006, 0.009] | 0.0072 [0.0069, 0.0076] |
| N | number | 7.1961 [7.1734, 7.2187] | 6.9031 [6.7617, 7.0459] |
| $\rho_c$ | day$^{-1}$ | $5.31 \times 10^{-9}$ [$3.76 \times 10^{-9}$, $7.12 \times 10^{-9}$] | $1.52 \times 10^{-8}$ [$1.47 \times 10^{-8}$, $1.58 \times 10^{-8}$] |
| $\mu_c$ s.t. $\mu_R^{(consti.)}(0) = \mu_c$ | molecules/cell | 7.3589 [7.1719, 7.5483] | 7.2479 [6.9544, 7.5475] |
| $\sigma_R^{(consti.)}(0)$ | molecules$^2$/cell | 0.4672 [0.4509, 0.4837] | 0.4841 [0.4748, 0.4935] |
| $\mu_s$ s.t. $\mu_R^{(synN)}(0) = \mu_s \frac{\mu_H^{n_H}}{\mu_H^{n_H} + K_H^{n_H}}$ | molecules/cell | 6.5367 [6.2714, 6.8075] | — |
| $\sigma_R^{(synN)}(0)$ | molecules$^2$/cell | 1.1257 [0.999, 1.2599] | — |
| $K_H$ | molecules/cell | $2.46 \times 10^5$ [$2.46 \times 10^5$, $2.46 \times 10^5$] | — |
| $n_H$ | number | 3.8795 [3.7879, 3.9722] | — |
| Fixed parameters | | | |
| $k_{off}$ (high-affinity CAR) | s$^{-1}$ | $9.0 \times 10^{-5}$ | $9.0 \times 10^{-5}$ |
| $k_{on}$ (high-affinity CAR) | nM$^{-1}$ s$^{-1}$ | $5.1136 \times 10^{-6}$ | $5.1136 \times 10^{-6}$ |
| $K_D$ (high-affinity CAR) | nM | 17.6 | 17.6 |
| $k_{off}$ (low-affinity CAR) | s$^{-1}$ | $6.8 \times 10^{-4}$ | $6.8 \times 10^{-4}$ |
| $k_{on}$ (low-affinity CAR) | nM$^{-1}$ s$^{-1}$ | $3.2381 \times 10^{-6}$ | $3.2381 \times 10^{-6}$ |
| $K_D$ (low-affinity CAR) | nM | 210 | 210 |

The 95% confidence intervals are shown in squared brackets.

differences in receptor clustering giving rise to immunological synapse formation at different CAR abundances (see the Discussion section).

Our PASCAR model allows us to analyze how CAR T cell subpopulations expressing different CAR abundances respond to target cells. We used Model KP with best fit parameters (Table 1) to carry out the analysis. The ability of a CAR T cell subpopulation expressing a specific CAR abundance to lyse target cells will depend on (i) the affinity ($k_{on}$, $k_{off}$) of CARs for HER2 and the strength of the ensuing signaling in the CAR T cell, and (ii) the number of member CAR T cells in the subpopulation. Therefore, a CAR T cell subpopulation expressing higher CAR abundances but containing a small number of member cells could lyse target cells at a lower rate than a subpopulation expressing lower CAR abundances with a larger number of member cells. We quantified the rate of lysis of target cells expressing abundances of HER2 antigens of magnitude $H$ by a CAR T cell subpopulation expressing $R$ number of CARs with $T_R$ number of member cells by $c_{RH}^{(lysis)} = \lambda_{RH} T_R$. The kinetics of $c_{RH}^{(lysis)}$ for target cells expressing mean $H$ abundance (or $\overline{H}$) shows an increase with time resulting from the increase in $T_R$ because of proliferation (Figs 3A and S3A–D). The CAR T cell subpopulations with intermediate range of CAR abundances ~(500–2,500 molecules/cell at t = 0) induce a larger rate of lysis of target cells compared with the subpopulations with higher or lower CAR expressions (Fig 3A). This is because of the following reason. The size of the subpopulation $T_R$ increases as $R$ increases from small to (~10 molecules/cell) to intermediate (~500 molecules/cell) and then starts decreasing as R increases further to larger values (~2,500 molecules/cell) (Fig S4A). In contrast, $\lambda_{RH}$ ($\propto R$, as $K_D \ll R$ or $K_D \ll H$) increases

monotonically with R (Fig S4B). However, the decrease in the subpopulation size (>10-fold) outweighs the increase (~fivefold) in $\lambda_{RH}$ as R increases from intermediate to larger values resulting in the decrease of $c_{R\overline{H}}^{(lysis)}$ (Fig S4C). The rate of proliferation of CAR T cell subpopulations as they interact with target cell subpopulations expressing specific abundances of HER2 antigens will depend on (i) the affinity ($k_{on}$, $k_{off}$) of CARs for HER2 and the strength of the ensuing signaling in the CAR T cell, and (ii) the number of target cells expressing a particular abundance (H) of the HER2 antigens. The proliferation rate of a CAR T cell subpopulation expressing abundances of magnitude $R$ and containing $T_R$ number of member cells as they interact with a target cell population of size $U_H$ expressing abundance of magnitude $H$ of HER antigens is quantified by $c_{RH}^{(prolif)} = \rho_{RH} U_H$. For a target cell subpopulation expressing mean abundance $\overline{H}$, $c_{R\overline{H}}^{(prolif)}$ is given by $c_{R\overline{H}}^{(prolif)} = \rho_{R\overline{H}} U_{\overline{H}}$. The proliferation rates of CAR T cell subpopulations increase with increasing abundances of CAR expressions (Figs 3B, S3, and S4) almost linearly with R because $\rho_{RH} \propto R$ when CAR and HER2 interact with high affinity, that is, $K_D \ll R$ and $K_D \ll H$. The initial CAR distribution estimated by our method follows a lognormal distribution, that is, the T cell subpopulation size for intermediate R is larger than that for larger R. However, the increase in the proliferation rate at larger CAR abundances is unable to increase the size of the subpopulation such that the population peaks at these large values of CAR abundances. In addition, the proliferation rate of the CAR T cells decreases with time as the number of target cells decreases over time because of the lysis of the target cells by the CAR T cells (Figs 3B, S3, and S4). Therefore, the peak of the population still remains at intermediate values of R values at later times.

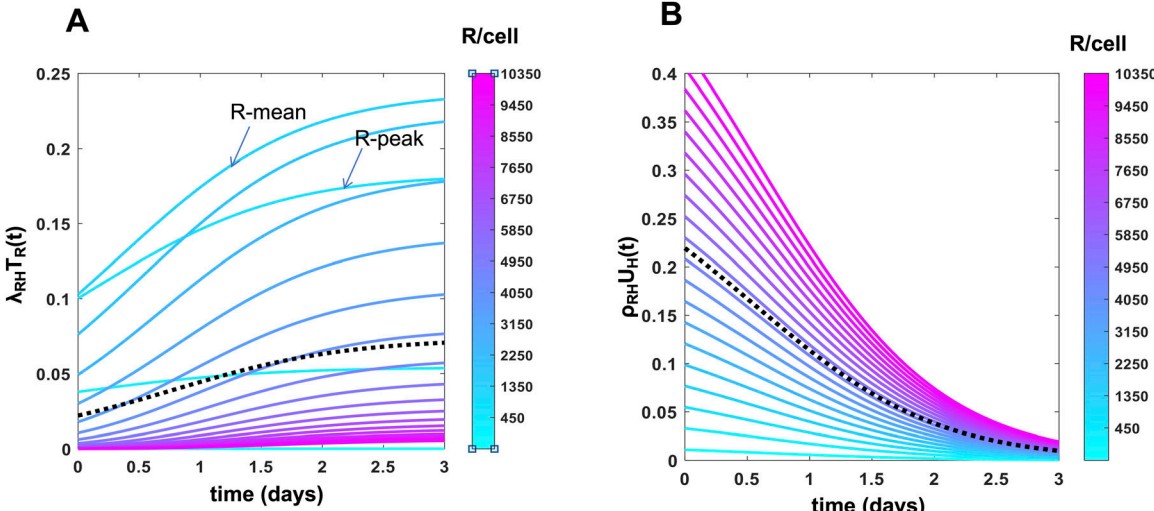

**Figure 3. Constitutive chimeric antigen receptor (CAR) T cell subpopulations expressing different CAR abundances show different cytotoxic and proliferative responses.**
**(A)** Shows lysis rates of target cells expressing mean human epidermal growth factor receptor 2 abundances (=$10^{6.2}$ molecules/cell) where the lysis is mediated by subpopulations of constitutive CAR T cells expressing different CAR abundances. The lysis rates increase with time as the numbers of cells in the CAR T cell subpopulations increase because of cell proliferation. The lysis rates for the CAR T cell subpopulations corresponding to the mean CAR expression (denoted as R-mean) and to the mode of the CAR distribution (denoted as R-peak) at day 3 are marked. The dotted line shows the average of lysis rates across the CAR T cell subpopulations. **(B)** Shows proliferation rates of CAR T cells as they interact with target cells expressing mean human epidermal growth factor receptor 2 abundances (=$10^{6.2}$ molecules/cell).

## Model with kinetic proofreading describes lysis of target cells by synNotch-CAR T cells

We applied Model KP to describe the cytotoxic response and proliferation of synNotch-CAR T cells when the CAR T cells were co-cultured with target cells in vitro. The mean CAR abundance of the synNotch-CAR T cells in Model KP was assumed to be proportional to a Hill function of the mean HER2 abundance of the target cells (details in the Materials and Methods section). We reasoned that CAR T cell signaling and activation in constitutive and synNotch-CAR T cells should involve the same physiochemical processes; therefore, we fitted the percentage lysis data (Hernandez-Lopez et al, 2021) and the CAR expression data (Hernandez-Lopez et al, 2021) for constitutive and synNotch-CAR T cells simultaneously with Model KP. Model KP fits the percentage lysis and the means and variances of CAR and HER2 abundances reasonably well (Fig 4A–C). The estimated parameters (Table 1) show a signaling cascade of N ≈ 7 best fit the data. This result is consistent with recent experiments (Britain et al, 2022) where clustering of LAT in T cell signaling is achieved in N = 7.8 ± 1.1 kinetic proofreading reaction steps. The estimated value of the phosphorylation rate, $k_p$ (≈0.007 s$^{-1}$) is larger than the ligand unbinding rate $k_{off}$ (≈$10^{-4}$ s$^{-1}$). The estimated Hill function parameters ($n_H$ ≈ 4, $K_H$ ≈ 2 × $10^5$ molecules/cell) imply that CAR expressions are induced sharply as HER2 abundances increase past 2 × $10^5$ molecules/cell giving rise to the almost binary cytotoxic response (off/on) against healthy cells (~$10^{4.5}$ HER2 molecules/cell) or tumor cells (>$10^{6.5}$ HER2 molecules/cell). Next, we tested the ability of Model KP to predict synNotch-CAR T cell response for experiments (Hernandez-Lopez et al, 2021) not included in training the model. We used Model KP to predict percentage lysis

of target cells by synNotch-CAR T cells at day 3 post-incubation where the synNotch-CAR T cells were co-incubated with different concentrations of target cells not included in model training. The model predictions captured the dependency of the cytotoxic response on the initial effector target ratio observed in experiments (Hernandez-Lopez et al, 2021) reasonably well (Fig 4D and E). We evaluated the correlations between the fitted parameters (Fig S5A) which showed low correlation (|r| < 0.5) between the model parameters; parameter $\rho_0$ and N showed the largest dependency (r = −0.49). We also predicted the variation of the percentage lysis for a cytotoxic assay performed at E:T ratio of 1:0.3 for target cells displaying different HER2 abundances interacting syn-Notch CAR T cells expressing the highest affinity CAR ($K_D$ = 1.9 nM, $k_{on}$ = 1.2 × $10^4$ M$^{-1}$ s$^{-1}$, $k_{off}$ = 2.2 × $10^{-5}$ s$^{-1}$) which showed excellent agreement (Fig 4F). We also computed the variations in the cost function (Equation (5)) with all pairs of parameters (Fig S5B). The analysis of the synNotch-CAR T cell subpopulations to induce cytotoxicity showed that similar to constitutive CAR T cells, synNotch-CAR T cells with an intermediate range of CAR abundances generated the larger response compared with the subsets with low and high CAR abundances (Figs 4G and S6A–D). The proliferation rates of synNotch-CAR T cell subpopulations increase with increasing CAR expressions (Figs 4H and S6A–D) similar to that of constitutive CAR T cells.

## Optimal design of constitutive and inducible CAR T cells

We performed Pareto optimization in the space of parameters that can be potentially manipulated in experiments. For constitutive CAR T cells, we carried out the analysis for CAR ligand-binding affinity parameters, $k_{on}$ and $k_{off}$, and, for synNotch-CAR T cells, in addition

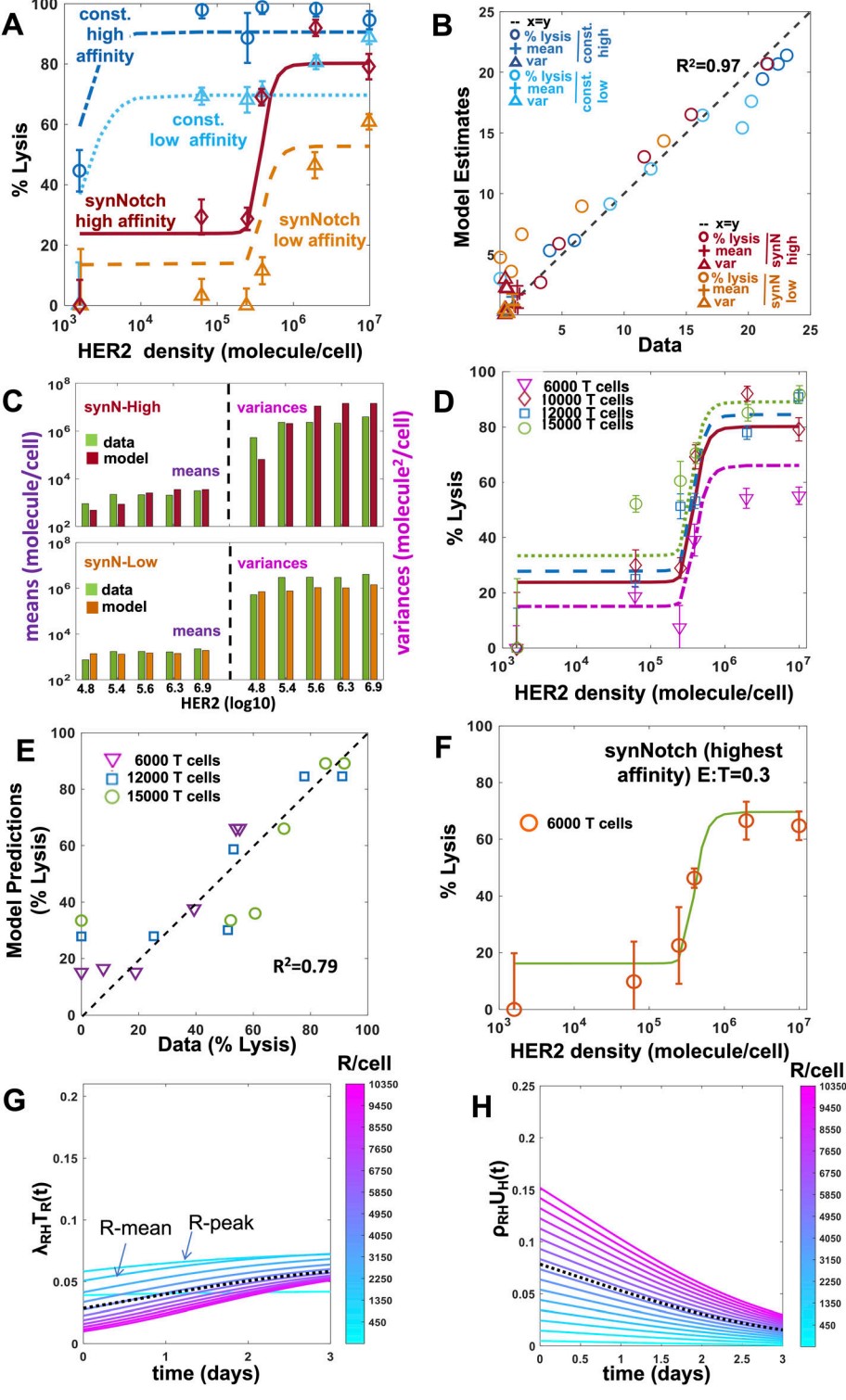

**Figure 4. Protein abundance structured population dynamic model for CAR T cell (PASCAR) modeling of cytotoxic and proliferation responses of constitutive and synthetic notch (synNotch) chimeric antigen receptor (CAR) T cell against target cells.**

**(A)** Shows fits for Model KP to the percentage lysis of target cells at 3 d after 10,000 synNotch or constitutive CAR T cells were incubated with 20,000 target cells. The data and the fits are shown for five different human epidermal growth factor receptor 2 (HER2) expressions with average HER2 abundances at $10^{4.7}$, $10^{5.2}$, $10^{5.7}$, $10^{6.2}$, $10^{6.9}$ molecules/cell. The comparisons of the distributions of CAR abundances between the model and the measured values at day 3 are shown in Fig S13. The synNotch and constitutive CAR T cells express either high affinity ($K_D$ = 17.6 nM, $k_{off}$ = 9.0 × $10^{-5}$ $s^{-1}$) or low affinity ($K_D$ = 210 nM, $k_{off}$ = 6.8 × $10^{-4}$ $s^{-1}$) CARs. **(B)** Comparison of the model fits for the percentage lysis, and means and variances of distributions of CAR abundances at day 3 post co-incubation with their experimental counterparts. The variables are made nondimensional by scaling the variables by their respective SDs calculated using the measured values. The goodness of fit is quantified by the correlation co-efficient $R^2$ which shows an excellent agreement ($R^2$ = 0.97) between the model and the data. **(C)** Comparison of the measured means and the variances of CAR abundances at day 3 post co-incubation with the fits obtained from Model KP. The mean HER2 abundances of the target cells used for modeling the co-culture experiments are shown along the x-axis. The means and variances for CAR abundances obtained from Hernandez-Lopez et al (2021) are shown with green bars. **(D)** PASCAR model predicted percentage lysis (solid and dashed lines) of target cells at 3 d after 6,000, 12,000, and 15,000 synNotch CAR T cells were incubated with target cells. The data for 10,000 synNotch CAR T cells were used for training the model which are also shown as reference. **(E)** The predictions in (D) are in good agreement ($R^2$ = 0.98) with the data obtained from Hernandez-Lopez et al (2021). **(F)** PASCAR model well-predicted percentage lysis (solid line) of target cells at 3 d for a cytotoxic assay at E:T = 0.3 where synNotch CAR T cells expressing high-affinity CAR ($K_D$ = 1.9 nM, $k_{on}$ = 1.2 × $10^4$ $M^{-1}$ $s^{-1}$, $k_{off}$ = 2.2 × $10^{-5}$ $s^{-1}$) were incubated with target cells. **(G)** Shows lysis rates of target cells expressing mean HER2 abundances (=$10^{6.2}$ molecules/cell) where the lysis is mediated by subpopulations of synNotch CAR T cells expressing different CAR abundances. **(H)** Shows proliferation rates of synNotch CAR T cells as they interact with target cells expressing mean HER2 abundances (=$10^{6.2}$ molecules/cell).

to the CAR affinity parameters, the threshold ($K_H$) and sharpness ($n_H$) of CAR expression, were also considered. To evaluate the optimal parameters, we set up in silico cytotoxicity assays, where a mixture of 20,000 healthy and tumor target cells with 10,000 constitutive or synNotch-CAR T cells were co-incubated in vitro

(Fig 5A). The CAR T cells and the target cells interacted following Model KP set at the best fit parameter values (Table 1) and the percentages of healthy and tumor cells lysed after 5 d of culture were computed. When synNotch-CAR T cells were present, we assumed the CAR expressions in the synNotch-CAR T cells are

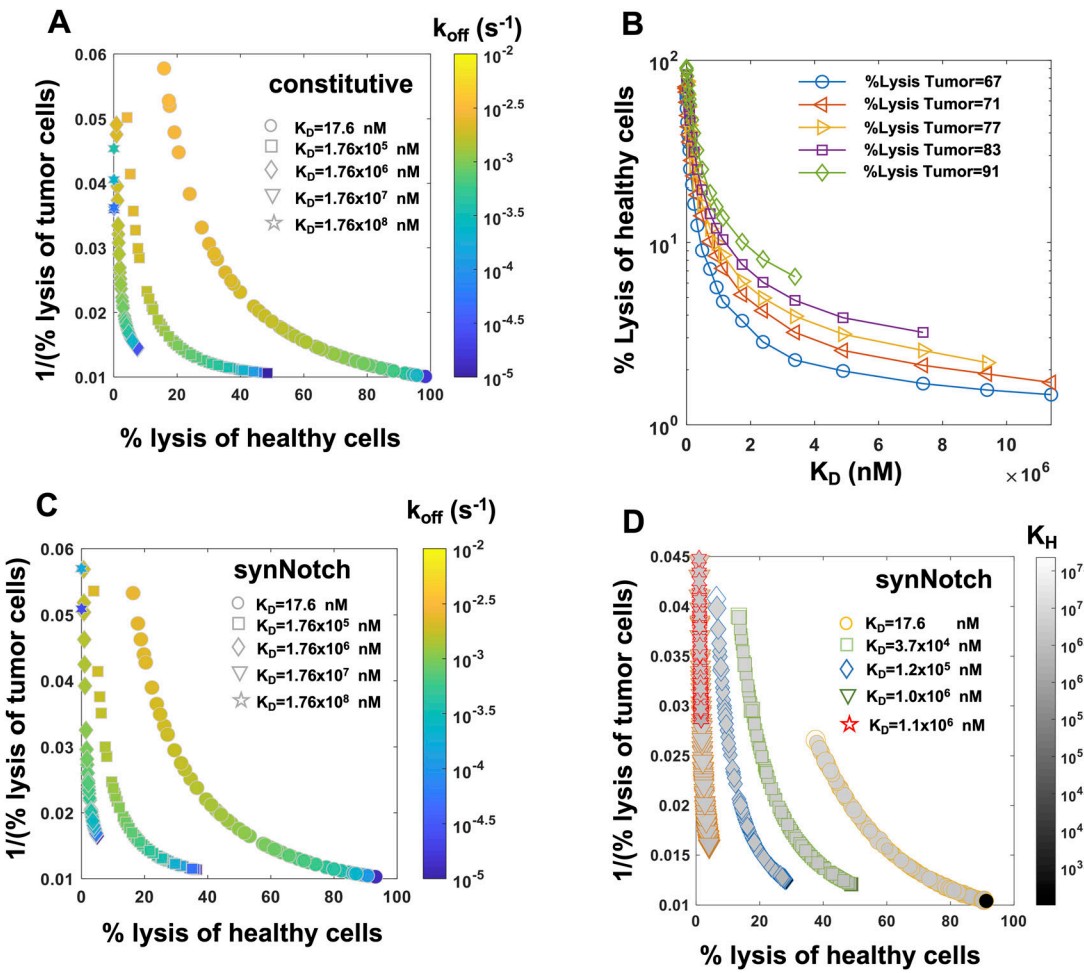

**Figure 5.   Pareto fronts revealing optimal responses of constitutive and synthetic notch (synNotch) chimeric antigen receptor (CAR) T cells against tumor and healthy cells.**
**(A)** Pareto fronts for constitutive CAR T cells in the plane spanned by % lysis of healthy cells and 1/(% lysis of tumor cells). The Pareto fonts are calculated for different dissociation constants $K_D = k_{off}/k_{on}$ where for each $K_D$ value, $k_{off}$ and $k_{on} = k_{off}/K_D$ are varied to obtain the corresponding front. The Pareto fronts are calculated 5 d after the CAR T cells were incubated with a 1:1 mixture of tumor and healthy cells (Fig S10) in silico (details in the main text) with tumor cell number being 20,000. **(B)** Variations of % lysis of healthy cells with $K_D$ for specific % lysis of tumor cells by constitutive CAR T cells along the Pareto fronts. The % lysis of healthy cells decrease with increasing $K_D$ until a certain value (the end points shown on the graph); increasing the $K_D$ further decreases % lysis of tumor cells as well (not shown on the graph). **(C)** Pareto fronts for synNotch CAR T cells for different dissociation constants $K_D = k_{off}/k_{on}$ where for each $K_D$ value, $k_{off}$ and $k_{on} = k_{off}/K_D$ are varied to obtain the corresponding front. The other parameters $K_H$ and $n_H$ are fixed throughout at $2.42 \times 10^5$ molecules/cell and 3.97, respectively. **(A)** The in silico cytotoxic assay is set up the same way as in (A). **(D)** Pareto fronts for synNotch CAR T cells for different fixed $k_{off}$ and $K_D$ values at ($9.0 \times 10^{-5}$ s$^{-1}$, 17.6 nM), ($5.0 \times 10^{-4}$ s$^{-1}$, $3.7 \times 10^4$ nM), ($5.0 \times 10^{-4}$ s$^{-1}$, $1.2 \times 10^5$ nM), ($7.0 \times 10^{-4}$ s$^{-1}$, $1.0 \times 10^6$ nM) and ($1.5 \times 10^{-3}$ s$^{-1}$, $1.1 \times 10^6$ nM), where $k_{on} = k_{off}/K_D$. $K_H$ and N are varied within ($1 \times 10^3$, $5 \times 10^7$) molecules/cell and (1, 8) on each Pareto front. The symbol size is proportional to N and the shades filling the symbols are proportional to $K_H$. **(A)** The in silico cytotoxic assays are set up the same way as in (A).

determined by the HER2 expressions of the tumor cells. The multi-objective optimization was performed in MATLAB where the competing objectives of percentage lysis of healthy cells and 1/(% lysis of tumor cells) were minimized simultaneously. The calculations of Pareto fronts showed that for constitutive CAR T cells, decreasing $k_{off}$ increases lysis of both healthy and tumor cells on a Pareto front at a fixed dissociation rate $K_D = k_{off}/k_{on}$. Decreasing $k_{off}$ increases the lifetime of the CAR–HER2 complexes allowing for the complexes to last longer than the waiting time required for the signaling reactions to generate the active signaling complex required to activate the CAR T cell. Thus, decreasing $k_{off}$ leads to increase in the destruction of tumor and healthy target cells.

However, Pareto fronts at different $K_D$ values revealed that increasing $K_D$ until a certain limit can increase the lysis of tumor cells, whereas decreasing healthy cell lysis (Fig 5A). For example, for a fixed % lysis of tumor cells (e.g., 66%), increasing $K_D$ can decrease % lysis of healthy cells from 60% to 20% (Fig 5A and B). However, increasing $K_D$ further starts decreasing lysis of tumor cells as well. This behavior can be explained by the dependency of the abundances of CAR–ligand complexes with $K_D$. Because the average abundances (~5,000 molecules/cell) of CARs are ~6 and ~600 times smaller than that of HER2 on tumor and healthy cells, respectively, most of the CARs in the CAR T cells form complexes with HER2 antigens displayed by the healthy and the tumor cells for high-

affinity CARs ($K_D \ll 5{,}000$ molecules/cell). As, $K_D$ is increased ($K_D >$ 5,000 molecules/cell), larger numbers of CAR-HER2 complexes are formed when CAR T cells interact with tumor cells compared with the healthy cells because of the availability of 100 times more HER2 antigens on tumor cells. However, as $K_D$ is increased further ($K_D \gg$ 5,000 molecules/cell), the copy numbers of CAR–HER2 complexes decrease substantially for CAR T cells interacting with tumor and healthy cells as the majority of the CARs are unable to form complexes because of the weak affinity of the binding (Fig S7). This results in inefficient lysis of healthy and tumor cells by CAR T cells. The above increase in discrimination between healthy and tumor cells with increasing $K_D$ upto a certain limit is qualitatively consistent with experiments (Caruso et al, 2015) with CAR T cells interacting with target cells expressing low (~30,899 molecules/cell [Caruso et al, 2015]) and high (~628,265 molecules/cell [Caruso et al, 2015]) EGFR abundances where the CAR was generated from high-affinity cetuximab ($K_D$ = 1.9 nM [Talavera et al, 2009]) or low-affinity nimotuzumab ($K_D$ = 21 nM [Talavera et al, 2009]). The experiments showed that the CAR T cells with nimotuzumab produced a lower killing of target cells with low EGFR abundances compared with the CAR T cells with cetuximab (Caruso et al, 2015). Both CAR T cells produced similar killing of target cells with high EGFR abundances (Caruso et al, 2015).

Next, we carried out our analysis for synNotch-CAR T cells. Similar to constitutive CAR T cells, the Pareto fronts for synNotch-CAR T cells for fixed $n_H$ and $K_H$ values showed increased tumor and decreased healthy cell lysis when $K_D$ increased for a fixed unbinding rate $k_{off}$ until a particular limit (Fig 5C). For a fixed HER2 affinity (fixed $k_{on}$ $k_{off}$), increasing $n_H$ and decreasing $K_H$ increased lysis of tumor and healthy cells (Fig 5D). This is because, increasing $n_H$ and decreasing $K_H$ produce higher CAR expressions in the synNotch-CAR T cells when they interact with tumor cells, and those CAR T cells induce greater lysis of tumor cells. However, the same CAR T cells become efficiently activated by healthy cells for higher affinity CARs ($K_D \ll 5{,}000$ molecules/cell, $k_{off} \sim 10^{-5}$ s$^{-1}$) and thus induce increased lysis of healthy cells. Thus, an optimal design of these inducible CAR T cells could be generation of CAR expressions with moderate affinities ($K_D \sim 1\ \mu M$, $k_{off} \sim 10^{-4}$ s$^{-1}$) with higher values of $n_H$ and lower values of $K_H$.

We also carried out the above Pareto front analysis for other ratios of healthy and tumor cells (1:4 and 4:1) for constitutive and syn-Notch CAR T cells which showed similar behavior (Figs S8A–I).

## Discussion

We developed a multiscale mechanistic model PASCAR that integrates processes across the scales of molecules of CARs and antigens to the populations of CAR T- and target cells. This multiscale approach is particularly relevant for modeling response of heterogeneous populations of CAR T cells expressing a wide range of CAR abundances against target cells displaying antigen abundances that can vary over 1,000-fold across target cells. We pursued an approximate or coarse-grained modeling approach where many microscopic details of CAR T cell signaling and activation were incorporated implicitly in model parameters. This reduced

approach allowed us to quantitatively explore the roles of CAR and HER2 (CAR antigen) abundances, CAR affinities, and the ensuing cell signaling in regulating CAR T cell responses against healthy and tumor cells. The ability of this approach to describe the percentage lysis data and the proliferation of T cell subpopulations with different CAR abundances in cytotoxic assays and to predict results outside model training shows the success of our approach in modeling antitumor responses of constitutive and inducible CAR T cells. We were also able to estimate model parameters reasonably well using flow cytometry measurements of CAR and HER2 expressions and percentage lysis data from cytotoxic assays. Therefore, the PASCAR model combined with data from standard immunoassays can be used to make quantitative predictions for various experimental conditions investigating CAR T cell responses in vitro.

Application of PASCAR to describe cytotoxic responses of constitutive CAR T cells showed the relevance of kinetic proofreading in CAR T cell signaling in discriminating response between high- and low-affinity CARs. Model NKP which did not contain any kinetic proofreading was unable to separate the cytotoxic responses mediated by high- ($K_D$ = 17.6 nM, $k_{off}$ = $9.0 \times 10^{-5}$ s$^{-1}$) and low-affinity ($K_D$ = 210 nM, $k_{off}$ = $6.8 \times 10^{-4}$ s$^{-1}$) CARs. Including kinetic proofreading steps at early time signaling events in time scales of $1/k_p$ (~2.4 min) in Model KP was able to separate the cytotoxic responses. There are several differences in signaling events induced by TCR and CAR stimulation, for example, LAT is weakly phosphorylated in CAR T cells (Salter et al, 2021), whereas LAT is robustly phosphorylated and forms condensates in TCR signaling within 40 s (McAffee et al, 2022). Therefore, signaling proteins that could correspond to the active complex in Model KP could represent different signaling proteins in TCR and CAR signaling, for example, experiments with light-gated immunoreceptors show the active complex corresponding the $N \approx 7.8$ in TCR signaling could represent LAT (Britain et al, 2022); however, the active complex in Model KP for $N$ = 7 is likely to represent a different signaling protein. Formation of immunological synapse where CARs along with the cognate ligands and associated siganling molecules spatially cluster at the interface of CAR T cell and the target cell plays an important role in CAR T cell activation. The degree of spatial clustering of these proteins can influence signal transduction and be affected by the CAR abundances and affinities. This could be a reason why our model underpredicted percentage lysis for the constitutive CAR T cells (Fig 2G) at low and high CAR abundances where PASCAR used the same signaling rates estimated for intermediate CAR abundances. A potential extension of PASCAR could be to include such details of signaling and spatial clustering that can be developed by building on several existing CAR T cell (Rohrs et al, 2018) and signaling models in lymphocytes (Čemerski et al, 2008; Grewal & Das, 2022).

We carried out a multi-objective optimization that minimizes destruction of healthy cells, whereas maximizing the elimination of tumor cells when a select set of model parameters that can be manipulated in experiments were allowed to vary. The calculation of Pareto fronts in our multi-objective optimization showed that intermediate values of CAR affinities led to an increase in tumor cell killing, whereas decreasing healthy cell killing. This finding is supported in previous experiments which showed reduction of CAR affinity reduced healthy cell killing but increased tumor cell killing

(Caruso et al, 2015). For inducible CAR T cells, the increase in the sharpness of CAR up-regulation in conjunction with intermediate range of CAR affinities can produce a desirable amount of tumor cell and healthy cell killing.

An exciting extension of PASCAR would be to describe CAR T cell response in vivo. CAR T cells undergo a maturation process over a longer duration (~weeks) than that considered here to give rise to exhausted (Good et al, 2021) and memory phenotypes (Ghorashian et al, 2019; Wilson et al, 2021 Preprint). Cytokines and chemokines in the tumor microenvironment contribute to this maturation process (De Boer et al, 2001; Bell & Gottschalk, 2021). These processes have been modeled for T cells (Carbo et al, 2013) and CAR T cells (Liu et al, 2021) using mechanistic or data-driven models which could be potentially incorporated in the PASCAR model.

## Limitations

The PASCAR model approximates signaling using a minimal model which might not be able to describe engineered CAR adaptors that manipulates the number of ITAMs (Majzner et al, 2020), or engineered T cells where specific signaling proteins such as RasGTPase-activating protein (Carnevale et al, 2022) are knocked out. However, some of these changes could be potentially included in our minimal signaling network by implementing signaling models that have described similar effects in other contexts (Das et al, 2009). The kinetics of the induction and the decay (Roybal et al, 2016) of synNotch CARs are not considered in the current model which could be relevant to describe responses of synNotch CAR T cells as they transit from microenvironments rich in tumor cells to healthy cells in vivo. An extension of the current model to include different compartments with explicit kinetics of synNotch CAR abundances could potentially address this issue.

PASCAR assumes that the CAR abundances in single $CD8^+$ T cells do not mix as they divide, that is, daughter cells have the same CAR abundances as the mother cell. However, proteins in human cells (e.g., H1299 non-small cell lung carcinoma cell line) have been observed to mix because of cell division (Sigal et al, 2006) in time scales longer than two cell generations. It is unclear if the CARs follow the similar pattern as the $CD8^+$ T cells proliferate. The doubling time scale for the faster proliferating $CD8^+$ T cells in our model is ~1.7 d and the mean doubling time of the $CD8^+$ T cell population is ~3 d; therefore, there will a negligible amount of mixing in the system because of cell proliferation if a similar mixing time scale as in Sigal et al (2006) occurs for the CAR $CD8^+$ T cells.

PASCAR also assumes that the target and the T cells are well mixed. In vitro cytotoxicity experiments are carried out in culture wells and for the experiments in Hernandez-Lopez we estimated 99.8% of the T cells were partnered with target cells (details in Supplemental Data 3). However, depending on the number of target and T cells, the number of target cells in the immediate vicinity of a T cell is likely to be varied (details in Supplemental Data 3), therefore, a weighted sum for the target and T cells in Equations (3), (4), and (5) would be more appropriate. We plan to include that in a future study.

# Materials and Methods

## Solution of the ODEs

We set up a rectangular lattice for the abundances $R$ and $H$ with lattice constants $\Delta_R$ and $\Delta_H$, respectively. $T_R$ and $U_H$ in the ODEs in Equations (3) and (4) denote the numbers of CAR T- and tumor cells with $R$ and $H$ abundances between $R$ to $R + \Delta_R$ and $H$ to $H + \Delta_H$, respectively. The ranges of R, $\Delta_R$, H, and $\Delta_H$, are chosen based on the ranges ($[R_{min}, R_{max}]$ and $[H_{min}, H_{max}]$) and the distributions of R and H measured in flow cytometry experiments in Hernandez-Lopez et al (2021). Given the parameters, $k_{on}$, $k_{off}$, $k_p$, $\lambda_c$, $\rho_c$, and the initial distributions of $R$ and $H$, the nonlinear system of ODEs is solved numerically in MATLAB using the ode45 function with Runge-Kutta 4 numerical method. The units of $R$ and $H$ in the ODEs are in (# of molecules)/cell.

## Unit conversion of kinetic rates

The rates $k_{on}$, $k_{off}$, and $K_D$ are usually provided in the literature in units of $(nM)^{-1} s^{-1}$ (or $(\mu M)^{-1} s^{-1}$), $s^{-1}$, and nM (or $\mu M$), respectively. We convert $k_{on}$ and $K_D$ rates into units of (# of molecules)$^{-1} s^{-1}$ and (# of molecules) to use in the ODEs where the units for CAR and HER2 abundances are given by (# of molecules)/cell. The unit conversion is carried out as follows. The nM unit is changed to (# of molecules)/$(\mu m)^3$ using 1 nM = 600 × $10^{-3}$ (# of molecules)/$(\mu m)^3$ = 0.6 (# of molecules)/$(\mu m)^3$. We assume CAR and HER2 molecules form complexes when these molecules are separated by a distance (d) of 2 nm (Helm et al, 1991) or smaller, and the CAR and HER2 molecules interact in the immunological synapse formed at the interface of the interacting CAR T cell and the target cell. The area (A) of the synapse region is taken as 1/2 times the area of a T cell (Teague et al, 1993) = ½ × 4π $(7/2)^2$ $\mu m^2$. The number of CAR and HER2 molecules present in the area $A$ is assumed to be half of the total numbers of these molecules present in individual cells. The parameters $K_D$ and $k_{on}$ in the units of (# of molecules) and (# of molecules)$^{-1} s^{-1}$ are obtained by the following relations: $K_D$ [# of molecules] = $K_D$ [(# of molecules)/$(\mu m)^3$]/$(Ad)$[$(\mu m)^3$] and $k_{on}$ [(# of molecules)$^{-1} s^{-1}$] = $k_{on}$ [(# of molecules)$^{-1} s^{-1}/(\mu m)^3$]/$(Ad)$ [$(\mu m^3)$].

## Parameter estimation

We estimated model parameters $k_p$, $\lambda_c$, $\rho_c$, and the mean ($\mu_R(0)$) and the variance ($\sigma_R(0)$) of CAR abundances at t = 0 for constitutive and synNotch CAR T cells by fitting percentage lysis data and means and variances of CAR T cells obtained at day 3 post-incubation with target cells in cytotoxic assays. For synNotch-CAR T cells, additional parameters describing up-regulation of CARs because of binding of synNotch receptors with HER2 ligands on target cells were estimated. We minimized the cost function (Wu et al, 2022 Preprint) below using levenberg-marquardt algorithm in MATLAB.

$$Cost\ function = \sum_{all\ experimental\ conditions\ (e.g.,HER2\ abundances)} \frac{\left((\% \ lysis)_{expt} - (\% lysis)_{model}\right)^2}{\eta^2_{(\%lysis)}}$$

$$+ \frac{\left((\mu_R)_{expt} - (\mu_R)_{model}\right)^2}{\eta^2_{(R)}} + \frac{\left((\sigma^2_R)_{expt} - (\sigma^2_R)_{model}\right)^2}{\eta^2_{(R^2)}}$$

$$(5)$$

where $\mu_R$ and $\sigma_R$ denote the mean and the SD of the CAR abundances at day 3 post incubation. $\eta^2_{(\%lysis)}$ denotes the variance in the % lysis in the experiments which were calculated from Hernandez-Lopez et al (2021) (Fig 2A in that reference). $\eta^2_{(R)}$ and $\eta^2_{(R^2)}$ denote the variance and the fourth cumulant in CAR expressions, respectively, which were calculated from Hernandez-Lopez et al (2021). The confidence intervals were estimated by the *nlparci* function in Matlab using the residuals and covariance matrices given by the *nlinfit* in Matlab.

We provide specific details of parameter estimation for constitutive CAR and synNotch-CAR T cells below. (1) *Constitutive CAR T cells*: The data are obtained from Hernandez-Lopez et al (2021) where human CD8+ T cells were engineered to express CARs constitutively that bind HER2 with high ($K_D$ = 17.6 nM) and low ($K_D$ = 210 nM) affinities. In their experiments, human leukemia K562 cell lines were engineered to express five different average concentrations ($10^{4.2}$, $10^{5.2}$, $10^{5.7}$, $10^{6.2}$, $10^{6.7}$ molecules/cell) of HER2 molecules which were used as target cells in cytotoxic assays. We fitted (*nlinfit* function in MATLAB) flow cytometry data (Fig S9, and Fig 1C in Hernandez-Lopez et al [2021]) with log-normal distributions to estimate means and variances of HER2 in target cells used in our ODE models. Similarly, the distributions of CAR abundances at day 3 post co-incubation were obtained by fitting the flow cytometry data (Fig S1C in Hernandez-Lopez et al [2021]) for constitutive CARs with log-normal distributions. We assumed the same CAR distributions for high- and low-affinity CARs. Means and variances for the CAR abundances at day 3 in the co-culture experiments were calculated from the estimated log-normal distributions. (2) *synNotch-CAR T cells*: Hernandez-Lopez et al (2021) developed tunable CAR T cells by engineering a synNotch receptor in CD8+ T cells. The synNotch receptor binds with HER2 on target cells with low affinity (210 nM) and acts as a high-density antigen filter for inducing CAR expressions in the T cells. The generation of CARs by the synNotch circuit was modeled implicitly. We assume that the changes in the CAR expression in the syn-Notch CAR T cells occur at a faster time scale than that of target cell lysis and T cell proliferation. Thus, the mean CAR abundance ($\mu_R$ (0)) in synNotch-CAR T cells at the start of a co-culture experiment is assumed to be a Hill function of the mean HER2 abundance ($\mu_H$) of the target cells given by $\mu_R(0) = \mu_S \frac{(\mu_H)^{n_H}}{(\mu_H)^{n_H} + (K_H)^{n_H}}$. $n_H$ is the Hill coefficient and for $n_H \geq 2$, $\mu_R$ (0) changes in a switch like all or none fashion as $\mu_H$ increases beyond the threshold $K_H$. The parameters $\mu_s$, $n_H$, and $K_H$ were estimated by fitting the percentage lysis (Fig 2A in Hernandez-Lopez et al [2021]) and the CAR expression data (Fig S1C in Hernandez-Lopez et al [2021]) obtained at 3 d after co-incubating target cells and CAR T cells in

cytotoxic assays. The estimated distributions of HER2 in target cells described for constitutive CAR T cells were used for modeling experiments with synNotch-CAR T cells as well.

**Pareto optimization**

A target cell population of 20,000 cells composed of a 1:1 mixture of healthy and tumor cells was taken at t = 0. The healthy and tumor cells expressed HER2 molecules following a linear super-position of two lognormal distributions where mean values of the HER2 molecules were set to $10^{4.5}$ molecules/cell and $10^{6.9}$ molecules/cell for the healthy and tumor cells, respectively. The SDs of each lognormal distribution were set to 0.3 to avoid any substantial overlap between the distributions (Fig S10). In our in silico cytotoxicity assays, we considered the above 20,000 target cells were mixed with 10,000 constitutive or synNotch CAR T cells at t = 0 expressing a high-affinity CAR ($K_D$ = 17.6 nM, $k_{off}$ = 9 × $10^{-5}$ s$^{-1}$). The distribution of CARs for the constitutive CAR T cells was constructed following lognormal distributions with the parameters estimated in Table 1. For the synNotch CAR T cells, we assumed all the 10,000 CAR T cells expressed CARs following a lognormal distribution with a mean value = $\mu_S \frac{\mu_H^{n_H}}{\mu_H^{n_H} + K_H^{n_H}}$ where $\mu_H$ = $10^{6.9}$ molecules/cell. The values of $n_H$ and $K_H$ were set to the values estimated in Table 1 or varied for the Pareto front calculations. We used *gamultiobj* routine in Matlab to compute the Pareto front by optimizing two conflicting objective functions: $f_1$ = % lysis of healthy cells at day 5, and $f_2$ = 1/(% lysis of the tumor cells) at day 5. The Pareto fronts were calculated for fixed $K_D$ values where $k_{off}$ were varied. These calculations (Fig 5C) for the synNotch CAR T cells fixed the values of $n_H$ and $K_H$ at the values shown in Table 1. For these calculations, we varied $f_1$ and $f_2$ as functions of $k_{off}$ within ranges (1.0 × $10^{-5}$, 1.0 × $10^{-2}$) s$^{-1}$ such that $k_{on}$ = $k_{off}$/$K_D$ for different values of $K_D$ fixed at (17.6, 1.76 × $10^5$, 1.76 × $10^6$, 1.76 × $10^7$, 1.76 × $10^8$) nM with all the other parameters fixed at the estimated values given in Table 1 for constitutive and synNotch CAR T cells.

Another set of Pareto fronts were calculated for synNotch CAR T cells (Fig 5D) where $n_H$ and $K_H$ values were varied and all the other parameters including $K_D$ and $k_{off}$ were fixed. In this case, $f_1$ and $f_2$ varied with $n_H$ and $K_H$ within ranges (1, 8) and (1 × $10^3$, 5 × $10^7$), respectively, with $k_{off}$ and $K_D$ values fixed at (9.0 × $10^{-5}$ s$^{-1}$, 17.6 nM), (5.0 × $10^{-4}$ s$^{-1}$, 3.7 × $10^4$ nM), (5.0 × $10^{-4}$ s$^{-1}$, 1.2 × $10^5$ nM), (7.0 × $10^{-4}$ s$^{-1}$, 1.0 × $10^6$ nM), and (1.5 × $10^{-3}$ s$^{-1}$, 1.1 × $10^6$ nM).

We computed the Pearson's correlation coefficients between the parameter values as follows. We estimated the model parameters for the SynNotch CAR T cells in 10,000 MCMC simulations and sampled the % lysis data with replacement. We then computed the mean % lysis and the SD (Fig S5). The MATLAB function *corrcoef* was used to compute the Pearson's correlation coefficients and the *P*-values.

# Data Availability

The MATLAB codes and the data used are available at the GitHub link https://github.com/Harshana4532/CART_Project_Matlab_03Jan2023.

## Supplementary Information

## Acknowledgements

This work is supported by the Research Institute at the Nationwide Children's Hospital. We thank members of Das laboratory for discussions and feedback. We also thank Dr. Rogelio Hernandez-Lopez and Dr. Wendell Lim for providing clarification for the data shown in Hernandez-Lopez et al (2021).

### Author Contributions

H Rajakaruna: data curation, software, formal analysis, investigation, visualization, methodology, and writing—original draft, review, and editing.
M Desai: data curation, software, investigation, methodology, and writing—original draft.
J Das: conceptualization, resources, formal analysis, supervision, funding acquisition, investigation, methodology, project administration, and writing—original draft, review, and editing.

### Conflict of Interest Statement

The authors declare that they have no conflict of interest.

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
