## [Reviewer comments · Life Science Alliance]

Life Science Alliance

PASCAR: A Multiscale Framework to Explore the Design Space of Constitutive and Inducible CAR T cells

Harshana Rajakaruna, Milie Desai and Jayajit Das

DOI: <https://doi.org/10.26508/lsa.202302171>

Corresponding author(s): Prof. Jayajit Das (The Ohio State University)

Review Timeline:

Submission Date:	2023-05-18
Editorial Decision:	2023-06-09
Revision Received:	2023-06-16
Editorial Decision:	2023-06-20
Revision Received:	2023-07-08
Accepted:	2023-07-12

Transaction Report:

Please note that the manuscript was reviewed at *Review Commons* and these reports were taken into account in the decision-making process at *Life Science Alliance*.

Review
COMMONS

June 9, 2023

Re: Life Science Alliance manuscript #LSA-2023-02171-T

Jayajit Das
Nationwide Children's Hospital and the Ohio State University

Dear Dr. Das,

Thank you for submitting your revised manuscript entitled "A Multiscale Protein Abundance Structured Population Kinetic Model Systematically Explores the Design Space of Constitutive and Inducible CAR-T cells" to Life Science Alliance. The manuscript has been seen by the original reviewers whose comments are appended below. While the reviewers continue to be overall positive about the work in terms of its suitability for Life Science Alliance, some important issues remain.

Our general policy is that papers are considered through only one revision cycle; however, given that the suggested changes are relatively minor, we are open to one additional short round of revision. Please note that I will expect to make a final decision without additional reviewer input upon re-submission.

Please submit the final revision within one month, along with a letter that includes a point by point response to the remaining reviewer comments.

To upload the revised version of your manuscript, please log in to your account: <https://lsa.msubmit.net/cgi-bin/main.plex>
You will be guided to complete the submission of your revised manuscript and to fill in all necessary information.

B. MANUSCRIPT ORGANIZATION AND FORMATTING:

Sincerely,

Reviewer #1 (Comments to the Authors (Required)):

The authors have addressed my concerns.

Reviewer #2 (Comments to the Authors (Required)):

The authors have added substantial modifications to their initial submission. However, some major issues raised in the first review were not appropriately addressed. Some of the remaining issues are listed below.

A proper quantification of model performances and a statistical comparison between models has not been performed. The authors use a R^2 score with a normalization using the experimental standard deviations to quantify model performances. They find similar scores for the NKP and KP models (0.96 and 0.97 respectively). Hence, their conclusion that the NKP model cannot account for experimental data is not supported. Model should be compared using the AIC score or Bayes factors.

Some modifications were not highlighted in the resubmitted manuscript. For instance, the numbers of target and CAR T cells (100000 target cells and 15000 CAR T cells in the original submission) have been corrected to match those used in the experimental work by Hernandez-Lopez et al. as suggested in the previous review. However, this correction is not highlighted in the resubmitted manuscript. Intriguingly, the authors seem to consider that these corrected values were those used in the original version of the manuscript.

The value of the binding rate k_{on} has a strong impact on the model behavior. With the value used by the authors, CAR receptors are saturated even for low number of HER2 ligands. The regime of non saturated receptors should be explored by varying the value of k_{on} and possibly by including k_{on} in the set of fitted parameters.

The analysis of the sensitivity of the model results (the "Cost" function) with respect to parameter values has yet to be done. The values of the "Cost" function should be plotted with respect to all pairs of parameter values.

The analytical expression for CN in the KP model is indeed correct but the parameter beta is not necessary. The expression of C0 for the NKP model could be re-used in the KP model. There is no need to include a supplementary file to detail the derivation of this classical result.

----- Response to Reviewers Comments -----

Reviewer #1 (Comments to the Authors (Required)):

The authors have addressed my concerns.

We are happy that the reviewer is satisfied with the revised manuscript. Thank you for all the constructive feedback.

Reviewer #2 (Comments to the Authors (Required)):

The authors have added substantial modifications to their initial submission. However, some major issues raised in the first review were not appropriately addressed. Some of the remaining issues are listed below.

We are happy that the reviewer found that the revisions improved the quality of the manuscript. Below we respond to the specific reviewers' comments. The changes in the manuscript are marked in green color text.

A proper quantification of model performances and a statistical comparison between models has not been performed. The authors use a R^2 score with a normalization using the experimental standard deviations to quantify model performances. They find similar scores for the NKP and KP models (0.96 and 0.97 respectively). Hence, their conclusion that the NKP model cannot account for experimental data is not supported. Model should be compared using the AIC score or Bayes factors.

Thank you for the suggestion. We computed the AIC for the NKP (AIC= 135.6606478) and the KP (AIC= 112.0529115) model which show that the KP model is favored by the data ($\Delta\text{AIC} \geq 13$). We have included the calculation in the revised manuscript.

Some modifications were not highlighted in the resubmitted manuscript. For instance, the numbers of target and CAR T cells (100000 target cells and 15000 CAR T cells in the original submission) have been corrected to match those used in the experimental work by Hernandez-Lopez et al. as suggested in the previous review. However, this correction is not highlighted in the resubmitted manuscript. Intriguingly, the authors seem to consider that these corrected values were those used in the original version of the manuscript.

We are not sure how about the source of the confusion. We had used the same sets of values for CAR T cells and the target cells in the original and the revised submissions. After we received first round comments from the reviewer we reached out to the main authors of Hernandez-Lopez to confirm if the values we used agree with the values used in experiments. Since there was no change in the values, we did not highlight those changes. Hope this settles the confusion.

The value of the binding rate k_{on} has a strong impact on the model behavior. With the value used by the authors, CAR receptors are saturated even for low number of HER2 ligands. The

regime of non saturated receptors should be explored by varying the value of k_{on} and possibly by including k_{on} in the set of fitted parameters.

We varied K_D from nanomolar to hundreds of micromolar for the Pareto front calculations. In these calculations k_{on} and k_{off} were varied for a fixed $K_D (=k_{off}/k_{on})$. We believe this explores the range of k_{on} the reviewer is asking for. For the fits to the data, the K_D and the k_{off} values were measured, which in turn fixes the k_{on} values in the experiments thus we kept these parameters fixed to the measured values. We do not believe fitting k_{on} values in micromolar ranges for these data is appropriate.

The analysis of the sensitivity of the model results (the "Cost" function) with respect to parameter values has yet to be done. The values of the "Cost" function should be plotted with respect to all pairs of parameter values.

We have included a figure in the supplement (Fig. S5B) where we show variations of the sum of squared residuals weighted by the standard deviations estimated from the experiments with all pairs of our model parameters. The fixed parameters for each case are set to the best fit values.

The analytical expression for CN in the KP model is indeed correct but the parameter beta is not necessary. The expression of C0 for the NKP model could be re-used in the KP model. There is no need to include a supplementary file to detail the derivation of this classical result.

We would be inclined to keep it in the supplementary materials as it might help junior trainees following the calculations in our manuscript.

--

June 20, 2023

RE: Life Science Alliance Manuscript #LSA-2023-02171-TR

Prof. Jayajit Das
The Ohio State University
Pediatrics
700 Childrens Drive
Columbus 43205

Dear Dr. Das,

Thank you for submitting your revised manuscript entitled "PASCAR: A Multiscale Framework to Explore the Design Space of Constitutive and Inducible CAR T cells". We would be happy to publish your paper in Life Science Alliance pending final revisions necessary to meet our formatting guidelines.

- please remove all unnecessary (old) files and only keep the ones mentioned for publication.
- please upload all your figures as single files; these will be displayed in-line in the HTML version of your paper, so please provide them as single page files (Figures); we do not have a limit on the number of main figures, and these can be split if necessary for space
- please add the Twitter handle of your host institute/organization as well as your own or/and one of the authors in our system
- please add an Author Contributions section to your main manuscript text
- please add a conflict of interest statement to your main manuscript text
- the legend of Figure S5 needs to be revised as no panels are present in the figure. Please correct callouts in the manuscript text accordingly
- please review and update the callouts for Figure S1 throughout the text
- please add callouts for Figures 4D, E,H ; S2A-B; S3A-D; S4A-C; S6A-D; S8A-I; S10 A-D; S11A-B; S13 to your main manuscript text
- please incorporate the Supplemental Material Text 1-3 into the main Materials and Methods section

A. FINAL FILES:

B. MANUSCRIPT ORGANIZATION AND FORMATTING:

Sincerely,

July 12, 2023

RE: Life Science Alliance Manuscript #LSA-2023-02171-TRR

Prof. Jayajit Das
The Ohio State University
Pediatrics
700 Childrens Drive
Columbus 43205

Dear Dr. Das,

Thank you for submitting your Methods entitled "PASCAR: A Multiscale Framework to Explore the Design Space of Constitutive and Inducible CAR T cells". It is a pleasure to let you know that your manuscript is now accepted for publication in Life Science Alliance. Congratulations on this interesting work.

DISTRIBUTION OF MATERIALS:

Again, congratulations on a very nice paper. I hope you found the review process to be constructive and are pleased with how the manuscript was handled editorially. We look forward to future exciting submissions from your lab.

Sincerely,
